# Recognition of intrinsic values of sentient beings explains the sense of moral duty towards global nature conservation

**Tianxiang Lan**[1]*, **Neil Sinhababu**[2], **Luis Roman Carrasco**[1,3]

**1** Department of Biological Sciences, National University of Singapore, Singapore, Singapore, **2** Department of Philosophy, National University of Singapore, Singapore, Singapore, **3** Campus for Research Excellence and Technological Enterprise, Singapore, Singapore

☯ These authors contributed equally to this work.

* lan.tx@nus.edu.sg

## Abstract

Whether nature is valuable on its own (intrinsic values) or because of the benefits it provides to humans (instrumental values) has been a long-standing debate. The concept of relational values has been proposed as a solution to this supposed dichotomy, but the empirical validation of its intuitiveness remains limited. We experimentally assessed whether intrinsic/relational values of sentient beings/non-sentient beings/ecosystems better explain people's sense of moral duty towards global nature conservation for the future. Participants from a representative sample of the population of Singapore (n = 1508) were randomly allocated to two "the last human" scenarios. We found that the best predictor of such a sense of moral duty for future nature conservation is the recognition of the intrinsic values of sentient beings. Our results suggest that the concern for animal welfare may enhance rather than compete with the sense of moral duty towards nature conservation.

## 1. Introduction

Contemporary environmental problems such as climate change and the biodiversity crisis have been partly attributed to certain anthropocentric ideologies that understand nature as a mere vessel of resources, or an entity to be conquered and subdued [1]. These ideologies are believed to provide justifications for people to dominate and cause damage to nature without qualm [2, 3]. Since the mid-20th century, philosophers have proposed various alternative moral theories that do not regard nature exploitatively. Most of those theories in the earlier decades may be loosely categorised as non-anthropocentric ethics, the core of which is the recognition of the intrinsic values of nature [4]. Intrinsic values may be understood as values that an entity inherently possesses, unaffected by its relationship with or evaluation by other entities. The entities with intrinsic values are deemed moral patients, which means that their interests and rights are a matter of ethical concern in their own right [4]. This contrasts with the concept of "instrumental values", referring to the value of an entity that is merely derived from its use in serving humans' interests. Entities with only instrumental values do not enjoy moral patienthood.

---

**Data Availability Statement:** Institutional Ethics Review Board (IRB) approval for this study was obtained under the condition that data would not be shared publicly under the Singapore's Personal Data Protection Act. The ethics committee of the

National University of Singapore to which data requests can be sent is available at irb@nus.edu.sg.

**Funding:** YES - This research is supported by the National Research Foundation, Prime Minister's Office, Singapore under its Campus for Research Excellence and Technological Enterprise (CREATE) Programme (NRF2016-ITC001-013). The funders had no role in study design, data collection and analysis, decision to publish, or preparation of the manuscript.

**Competing interests:** The authors have declared that no competing interests exist.

Non-anthropocentric moral theories mainly differ on the criteria for ascribing moral patienthood to an entity. For example, sentientism argues that a being's ability to sense pleasure and pain (i.e. sentience) is the only criterion for having moral patienthood [5]. On the other hand, biocentrism argues that all living beings have intrinsic values, each of them has its interests and is an end in itself. Their interests lie in their life, health and the power to cope with the changing environment [6]. Biocentrism allows non-sentient beings like plants and fungi to have moral patienthood just like sentient beings, such as humans, do.

Both theories above fall into the category of individualism, by which only the individual living organisms are moral patients. This contrasts with holistic theories that argue that ecosystems also have moral patienthood [1]. One of those theories is ecocentric holism which argues that biotic communities as holistic entities have intrinsic values, and that the "integrity, stability" [7, as cited in 11] or the "order" [8] of these communities is what should be conserved. The interest of the ecosystem is not reducible to the aggregate interests of the individual organisms in the community, and philosophers like Callicott [9, 10] even argued that the former should trump the latter. Proponents of holistic theories argue that such theories produce more intuitive judgments on modern environmental issues. Sagoff [11] argued that Singerian animal ethics necessarily obligates humans to interfere with nature to reduce pain in wild animals, which is inconsistent with ecologists' aim of conserving whole ecosystems. Similarly, biocentrism alone may also be understood as incapable of judging climate change as morally bad [12]. In contrast, holistic theories purportedly resonate better with the "evolutionary-ecological worldview" of modern environmentalists [10] and judge biodiversity crisis and climate change as morally bad in themselves.

## 2. The rise of relational values

The above debate goes on between merely recognising the instrumental values of nature versus extending intrinsic values to it. The debate usually assumes that anthropocentrism supports the instrumental exploitation of nature. In recent decades, however, this debate has drawn the criticism that it is obsessed with the imagined dichotomy between intrinsic and instrumental values [13]. It has been argued that intrinsic and instrumental values may not be the only two sources of justification for conservation. Instead, notions like sustainability, intergenerational equity and a sense of place [14, 15] may also be invoked to morally criticise humans' preferences for destructive economic development. Such justifications still peg the value of nature to humans' evaluation, but they advocate for the non-exploitative use of nature. Common to such criticism is the notion that only focusing on intrinsic or instrumental values ignores the present humans' relationships with the future generation, the cultural community, and the non-humans. The argument then goes that when making conservation decisions, such relationships should be considered as morally relevant as (if not more relevant than), say, the interests and rights of the individual humans or non-humans [16].

Building on the momentum of the above critique, the notion of "relational values" of nature as a source of justification for conservation has increasingly attracted the attention of conservationists. The term "relational values" is presumably first theorised and defined by Muraca [17] and later adopted and popularised by the Intergovernmental Science-Policy Platform on Biodiversity and Ecosystem Services (IPBES) [18]. The relational values framework stipulates that the value of nature may arise from its relationship with humans. When the relationship is purely that of exploiting-being exploited, nature has only instrumental values to humans. In such cases, humans have no direct moral duty towards nature. However, certain special relationships give rise to values that engender humans' direct moral duty towards nature [17]. Values arising from such special relationships are termed relational values (RV). Though

instrumental values were initially regarded as part of the RV, Himes and Muraca [19] later defined instrumental values as separate from RV, since instrumental values engender no direct moral duty towards the bearer of the value. This study thus follows the latest [19] interpretation and understands RV as values arising from non-instrumental relationships.

Muracian theory stipulates that two types of RV are relevant for humans' direct moral duty towards nature: "fundamental-relational" values and "intrinsic-eudaimonic" values. Fundamental-relational values arise from humans' "fundamental dependency" on nature. This special relationship of fundamental dependency is explained as nature forming "the totality of conditions for being" that "grounds the very possibility of a subject to occurring. . . and establishing specific, even instrumental relations to the world it inhabits" [17, 20]. Muraca further uses "atmosphere, photosynthesis, solar radiation. . . social, spiritual and cultural embedment" as examples of such essential conditions [17]. In other words, nature plays an indispensable role in producing and sustaining the conditions conducive for humans to not only survive and reproduce, but also to form meaningful social relationships, to search for and pursue one's sense of meaning of life and to form a sense of identity. This indispensable role of nature makes humans fundamentally dependent on nature.

Meanwhile, intrinsic-eudaimonic values arise from entities that provide "amenity values, aesthetic experiences" and also the "necessary condition for flourishing and leading a good human life". Here the explanation seemingly overlaps with the fundamental-relational concept. According to Muraca [17], relations that give rise to intrinsic-eudaimonic values are termed "functional relations". Compared to functional relations, the causal efficacy of fundamental relations is more "systemic, complex and process-like". We interpret functional relations as applicable to individual sites, beings, ecosystems that one directly loves, worships, or forms emotional bonds with. We interpret the fundamental relations as applicable to beings and systems that support, sustain and provide the context for such functional relations. In practice, however, one may not always have to maintain a clear distinction between the two special relationships.

In short, it can be gleaned from the above explanations that RV arise from humans' dependence (fundamental or functional) on nature for basic physical needs like sustenance, shelter and suitable climate; for basic psychological needs like identity formation and meaning-making; and for having a good life through forming meaningful social-cultural relationships, recreation, spiritual beliefs and so on. Such special relations are akin to humans' extraction of "ecosystem services" which also encompass physical, recreational, and social-cultural (spiritual) services. Yet the extraction of such services is not and should not be the purpose of one's care for nature; what motivates the care for nature is still its relationship with humans [19], distinguishing RV from instrumental values. Meanwhile, RV of nature are contingent upon the existence of humans' special (dependent) relationship with nature, unlike intrinsic values which do not change with how other entities relate to the value bearer. By grounding nature's value in its relationship with humans, RV theory may remain anthropocentric and yet advocate for the care for nature, thus resisting the assumption that anthropocentrism means instrumental exploitation of nature.

RV received substantial attention given the inclusion of this concept in the IPBES framework [18]. RV have however remained largely theoretical, with limited empirical validation [21]. Given the rapid inclusion of RV in global and national policies, it is relevant to investigate whether RV explain individuals' sense of duty towards nature conservation for the future, and the concern for which aspects of nature better explains this sense of duty. The latter is useful as the Muracian framework does not adjudicate between sentientism and biocentrism. To fill these research gaps, we intend to investigate the following research questions:

- Does recognising the RV of nature explain people's sense of moral duty towards global nature conservation for the future?

- For people who believe that humans have a moral duty towards global nature conservation for the future, which is the best predictor of such moral intuition–their recognition of the intrinsic values vs. relational values and sentient beings vs. non-sentient beings vs. ecosystems?

We derive two hypotheses based on the two research questions above:

H1: Recognising the RV of nature explains people's sense of moral duty towards global nature conservation for the future.

H2: Amongst the six potential predictors of the sense of moral duty towards global nature conservation for the future (the recognition of the intrinsic values vs. relational values and sentient beings vs. non-sentient beings vs. ecosystems), the two most important predictors of this moral intuition are their recognition of ① the RV of the ecosystem and ② the intrinsic values of sentient beings.

Note that in this article, the word "nature" refers to the non-human environment in general, whether understood individualistically or holistically. The word "conservation" refers to a generic sense of wanting to benefit, protect, or reduce and mitigate one's impacts on nature, to avoid the debate of what constitutes legitimate or productive conservation efforts.

## 3. Disambiguation of terms

First, the term "relational values" requires clarification. Stålhammar [22] pointed out two ideas surrounding RV in environmental ethics, and we interpret them as the weaker and the stronger conception of RV respectively. The first, weaker idea is that values are present not (only) in things but also in the relation, that the relation is more than just a means to serve the ends of the beings in that relation. This idea is consistent with the previous explanation that RV is contingent upon the special relation between humans and nature. The second, stronger idea argues not only that values lie in relations, but also that humans and nature are always in this special relation. According to this stronger idea, the tradition of seeing nature as exterior to the society should be rejected. The social and the ecological is believed to be so intertwined that it may not make sense to regard them as separate or separable. The relation between humans and nature is said to precede the distinction between the observer and the observed [17, 23], which means this relation is *necessarily* present. Note that in Muraca's original work [17], this necessarily present relation is said to be between humans and the world. Nevertheless, based on the feedback we received from other philosophers, many supporters of RV seem to equate "the world" with "the natural environment"–i.e., humans are *always* in a special relation with nature. In this paper, we adopt the weaker conception of RV only. This is because by building the Earth's natural environment into the conception of human society, the stronger conception entails a more rigid understanding of human nature and commits to bolder assumptions. For example, under this concept, a society living in an entirely artificially created environment through, say, space exploration cannot possibly exist (or it fails to qualify as a "human" society). To avoid making such strong empirical and ontological assumptions, this paper interprets RV as value arising from humans' special relations with nature only, without presuming the necessity or contingency of such relations.

Second, the term "intrinsic values" also needs clarification. McShane [24] pointed out that this term may mean different ideas. For example, it could mean something being cared about for its own sake pertaining to the agent's attitude, i.e., the agent's action towards object A aims to promote A's own good, rather than to promote the good of another object B. This is termed

"final values". On some other occasions, it may refer to non-extrinsic values. This means values that something has in virtue of its intrinsic properties, "independently of any connection to any valuable thing" [22, 24]. Still, it may also refer to "non-instrumental values"–any values that are contrasted against instrumental values which are understood as values derived merely from serving humans' interests, i.e., being humans' "instrument". There are many more possible meanings of "intrinsic values" which are not listed here. This study interprets intrinsic values as non-extrinsic values, for two reasons. Theoretically, defining intrinsic values as values that exist independent of an entity's external relations (which has a moral realist overtone) contrasts well with RV which is dependent on an entity's relation with humans. We may thus be reasonably confident that the intrinsic values survey items developed based on this definition are measuring a construct sufficiently distinct from RV. Practically, it is also a broader concept compared to other definitions, and it overlaps very much with the notion of non-instrumental values that the term "intrinsic values" most commonly refers to–recall that RV are proposed for overcoming the "dichotomy" between intrinsic values and instrumental values, for instance. This means survey items developed based on this definition may better resonate with how the term is popularly used.

## 4. Method

### 4.1 Experimental moral philosophy

We used the method of experimental moral philosophy that may be understood as empirical research on people's intuitive moral judgment, usually under controlled conditions [25, 26]. This method contributes to the conceptual analysis in ethics and informs philosophers whether a proposed definition of a concept fits with how the concept is intuitively conceived of by people, in actual or hypothetical cases [25].

### 4.2 Data collection

The experiment was performed through an online market research company that conducted an English-language survey from 14 October 2020 to 2 November 2020.Two versions of the survey were prepared (the dependent and independent group) and the subjects were randomly presented with either version (survey questions are available in the supplementary information). The survey was approved by the Institutional Review Board (IRB) at the National University of Singapore (NUS-IRB-2020-083). Survey participation was voluntary and was limited to people who were at least 20 years old. Survey participation was also limited to Singaporean citizens and permanent residents. To incentivise participation, the market research company issues survey participation points to subjects who complete the survey. Written informed consent was obtained from the subjects.

The subjects were stratified according to age, gender, race, household income, and residential region, such that the percentage composition of each demographic group surveyed reflects that of the overall population in Singapore. The stratification was achieved through the demographic screening questions in the survey. Once the quota of a certain demographic group was full, no more subjects belonging to that group were engaged for full participation.

### 4.3 Experiment design and prediction

Subjects were presented with a questionnaire with multiple sections. Subjects rated how much they agreed with certain statements on a one (strongly disagree) to seven (strongly agree) scale. The statements reflected how much they acknowledged the intrinsic and RV of ecosystem,

**Table 1. Value statements used in the survey.**

| Target | Category | Statement |
|---|---|---|
| Ecosystem | Intrinsic | Nature* is valuable in itself, independent of anything else. |
| | | We should consider protecting nature, regardless of whether this benefits humans or not. |
| | Relational | I have a strong affinity towards nature and ecosystems which forms part of my sense of identity. |
| | | I find spirituality in nature. |
| Sentient beings | Intrinsic | Sentient organisms are valuable in themselves, independent of anything else. |
| | | We should consider protecting sentient organisms, regardless of whether this benefits humans or not. |
| | Relational | I have a strong affinity towards sentient organisms, which form part of my sense of identity. |
| | | I find spirituality in sentient organisms. |
| Non-sentient beings | Intrinsic | Non-sentient organisms are valuable in themselves, independent of anything else. |
| | | We should consider protecting non-sentient organisms, regardless of whether this benefits humans or not. |
| | Relational | I have a strong affinity towards non-sentient organisms, which form part of my sense of identity. |
| | | I find spirituality in non-sentient organisms. |

*The word 'nature' in this context is best understood as evoking a sense of the natural environment as a whole, i.e., as holistic ecosystems. This is different from the usage of the word 'nature' in the body paragraphs of the article which refers to the environment in general (whether understood individualistically or holistically).

sentient beings and non-sentient beings respectively (Table 1). The statements were taken from or inspired by previous literature [27–30].

Subjects were then randomly assigned to either a "dependent group" or an "independent group". In each group, they read the following scenarios inspired by the "last [hu]man" argument [31]:

> In year 2022, humans fought a devastating world war. All humans except one died, while other living things are unaffected. Before the war, humans **have been dependent/are no longer dependent** on nature for ecological services* like today. The last surviving human on Earth, "The Last Human" was about to die, and activated a device to destroy all living things on Earth in 1000 months. The Last Human passed away soon afterwards, and all the remaining life forms and ecosystems on the planet were indeed destroyed 1000 months later as planned. The non-living structure of the Earth remained intact.

Where either of the options of the text in bold was used in the dependent and independent scenario respectively. In the scenarios, the destruction was stipulated to happen long after The Last Human's death, to test people's moral intuition towards the future nature.

After being presented with the scenarios, the subjects were then asked to rate the extent to which they agree that the action of The Last Human was morally wrong, on a one to seven scale. The moral judgment on The Last Human's action may be understood as reflective of the subjects' sense of moral duty towards global nature conservation.

The subjects were also asked how much they agreed that humans were fully dependent on nature in real life. The response to this question was used to control for "unconscious realism" [32], whereby people judge a hypothetical scenario by substituting the assumptions in the story (in this case, humans becoming fully independent from nature) with their belief of what

would happen in real life instead. Therefore, the potential influence of someone's doubt about whether humans can survive without nature should be controlled by including the answers to the above question in the statistical model.

For the independent group, humans' dependent relationship with nature (which is a source of RV) was portrayed as absent. Nature (present or future) therefore had little RV in the story, and the recognition of RV should not explain the moral judgment of The Last Human's action. Compared with the moral judgment scores in the independent group, if the scores in the dependent group did not differ significantly from the former, it would suggest that the recognition of RV also does not explain the moral judgment in the dependent group. This then means the results do not support H1 that RV explain people's sense of moral duty towards the future nature. Similarly, compared to the correlation between average RV statement scores and moral judgment scores in the dependent group, if the correlation between the two in the independent group does not differ significantly from the former, it means the results do not support H1. Though not explicitly stated, the moral intuition here is presumably directed at the global environment rather than the local environment, since the last human scenario is presented in the context of a general imaginary world and the scenario outcome is also general global destruction. Similarly, the intrinsic and relational values items also elicit more of a global as opposed to local concern, since they are phrased in general terms and do not make salient the unique context of each participant's experience of nature.

## 4.4 Data analysis

To verify whether the subjects interpreted the RV as a distinct construct from the intrinsic values, we carried out a Confirmatory Factor Analysis (CFA) using Correlated Traits Correlated Methods (CTCM) by separating the statements into two "traits" (intrinsic and relational values) and three "methods" (ecosystem, sentient beings and non-sentient beings). The correlation among all five constructs was not constrained. The comparative fit index (CFI), normed fit index (NFI) and root mean square error of approximation (RMSEA) values were examined to assess the goodness-of-fit of the model. Configural invariance was tested by fitting the same model onto the dependent group and independent group. Internal consistency reliability was checked using Cronbach's alpha. The above steps of analysis were performed using the package ltm [33] and lavaan [34] in R [35].

To assess H1, Mann-Whitney U test was first performed on the moral judgment scores on The Last Human's action in the dependent group and the independent group to assess whether the average moral judgment scores of the dependent group were significantly higher than that of the independent group. Ordinal regression was then performed using the moral judgment scores (*MorJudg*) as the dependent variable. Moral judgment scores were regressed against the average scores for all the RV statements (*AveRE*), the respondents' assigned grouping (*DepIndep*), and their interaction. The process was repeated while adding the degree of agreement that humans were fully dependent on nature in real life (*HumanRelyNature*) as a control variable, together with its interaction term with the grouping variable, to control for unconscious realism. The two models assessed were:

Model 1: MorJudg~ AveRE+DepIndep+AveRE·DepIndep

Model 2: MorJudg~AveRE+DepIndep+HumanRelyNature+AveRE·DepIndep +DepIndep·HumanRelyNature

To assess H2, ordinal regression was used to regress moral judgment scores against the following six predictor variables: the average scores for statements of intrinsic values targeting ecosystem (*EcoIN*, Model 3), sentient organisms (*SenIN*, Model 4), and non-sentient organisms (*NSIN*, Model 5); the average scores for RV statements targeting ecosystem (*EcoRE*,

Model 6), sentient organisms (*SenRE*, Model 7), and non-sentient organisms (*NSRE*, Model 8) respectively. The six models were then compared based on their Akaike information criterion (AIC) value to assess which predictor variable correlated the best with the moral judgment. Given that some theorists argue that RV emphasises that knowledge and practices are situated in their particular contexts [22], we repeated the analyses while controlling for the demographic variables age and gender. Those variables may serve as a proxy for the contexts in which one constructs one's knowledge. The demographic variables were not significant and the results did not change. We also assessed whether the statement "I have a strong affinity towards ___ which forms part of my sense of identity" measures RV well by repeating the analyses after removing the affinity & identity statement for all three moral patient types. The results did not change. All the ordinal regressions above were performed using the MASS package in R [36].

## 5. Results

A total of 1508 responses were collected from the survey. 741 subjects were assigned to the dependent group and 767 subjects were in the independent group. 736 subjects were female and 772 subjects were male. The age of the subjects ranged from 21 to 80. The majority of the subjects were Singaporean citizens [1344 subjects).

The RV statements had a very high Cronbach alpha ($> 0.9$). CFA analysis yielded high CFI and NFI values ($>0.95$) and low RMSEA value ($<0.08$), suggesting good model fit. The same path was fitted onto the dependent and independent group through Multi-Group CFA, which also yielded high CFI and NFI value ($>0.95$) and low RMSEA value ($<0.08$), suggesting that Configural Invariance was met.

The mean value of the moral judgment scores of the dependent group was 5.372 while that for the independent group is 5.392. We failed to find evidence that the mean value in the dependent group was significantly higher when judging The Last Human's action as wrong compared to the independent group (Mann-Whitney U test p>0.05).

In Models 1 and 2, the confidence intervals for the experimental dependent-independent to nature treatment (DepIndep) and for the interaction term (AveRE:DepIndep) crossed zero and were not statistically significant (Tables 2 and 3).

Among Models 3–8, support for models that contained intrinsic values statements was clearly higher than models containing RV statements, with about 234 AIC units difference between the least supported intrinsic values model and the most supported RV model (Table 4). The most supported model corresponded to intrinsic values statements targeting sentient beings as the predictor variable (Table 4). When repeating the analysis after removing the affinity & identity statements, the AIC values of models for scores of RV statements increased slightly. The AIC ranking of the models did not change.

## 6. Discussion

The results of this study agree with the second part of hypothesis H2's prediction. The intrinsic values of sentient beings are the best predictor of people's sense of moral duty among the six

**Table 2. Output of ordinal regression Model 1.** AveRE: the average scores for all the RV statements. DepIndep: the respondent's assigned grouping.

| | Coefficient | 95% confidence interval | | t value |
|---|---|---|---|---|
| AveRE | 0.517 | 0.251 | 0.784 | 3.801 |
| DepIndep | -0.168 | -1.012 | 0.674 | -0.391 |
| AveRE:DepIndep | 0.034 | -0.135 | 0.202 | 0.391 |

**Table 3. Output of ordinal regression Model 2.** AveRE: the average scores for all the RV statements. DepIndep: the respondent's assigned grouping. HumanRelyNature: the scores for the statement that humans are fully dependent on nature in real life.

| | Coefficient | 95% confidence interval | | t value |
|---|---|---|---|---|
| AveRE | 0.142 | -0.153 | 0.438 | 0.941 |
| DepIndep | -0.339 | -1.369 | 0.691 | -0.646 |
| HumanRelyNature | 0.826 | 0.540 | 1.114 | 5.643 |
| AveRE:DepIndep | 0.077 | -0.108 | 0.263 | 0.816 |
| DepIndep:HumanRelyNature | -0.005 | -0.184 | 0.174 | -0.057 |

tested predictors. The results disagree with the first part of H2's prediction: the RV of the ecosystems are the 4th strongest predictor amongst the 6. The results also do not show evidence that the inclusion of RV through human dependence on nature in "the last human" story affected people's moral judgment score, which means hypothesis H1 that RV explain the sense of moral duty towards global nature conservation for the future was not supported.

At a practical level, the notion that recognising the intrinsic values of sentient beings is a major predictor of people's sense of moral duty towards global nature conservation for the future seems to weigh in favour of traditional sentientism. The fact that intrinsic values best predict the moral intuition to protect future nature means it is less likely that traditional sentientism distracts people from caring for nature [37]. Though empirical findings as such cannot offer theoretical protection for sentientism, they may provide a reason to be confident that belief in sentientism may work as well as, or even better than, holistic theories in explaining and eliciting laypeople's sense of moral duty to conserve nature. At the very least, the results provide a reason to doubt the intuitiveness among the public of arguments like "animal-welfare ethics have been either irrelevant to conservation and environmental concerns or an impediment to them", or "conservationists *qua* conservationists are simply not concerned about the welfare of individual organisms" [10].

Meanwhile, advocates of the intrinsic values of non-humans (individuals or collectives) may also revel in the finding that average scores for intrinsic values statements in general correlate with the moral judgment score much better than average scores for RV statements do. The results may lend confidence to philosophers who argue that intrinsic values play a crucial role in justifying moral concerns for nature that is irreplaceable by RV [38].

Our results may also inform the long-standing "new conservation" vs. "traditional conservation" debate, i.e., whether conservation should be justified by the benefits nature provides to humans ("new conservation") or because nature itself is valuable ("traditional conservation") [39]. In particular, the results may give the "traditional" camp more confidence in the effectiveness of appealing to intrinsic values to motivate people to protect nature, or at least the sense

**Table 4. Output of ordinal regression Model 3–8.** EcoIN: The average scores for statements of intrinsic values targeting ecosystem. SenIN: The average scores for all statements of intrinsic values targeting sentient organisms. NSIN: The average scores for all statements of intrinsic values targeting non-sentient organisms. EcoRE: The average scores for statements of RV targeting ecosystem. SenRE: The average scores for all statements of RV targeting sentient organisms. NSRE: The average scores for all statements of RV targeting non-sentient organisms.

| Predictor | Coefficient | 95% confidence interval | | t value | AIC (low to high) |
|---|---|---|---|---|---|
| SenIN | 1.071 | 0.973 | 1.170 | 21.32 | 4381 |
| NSIN | 0.965 | 0.872 | 1.060 | 20.11 | 4441 |
| EcoIN | 0.918 | 0.825 | 1.011 | 19.34 | 4486 |
| EcoRE | 0.546 | 0.464 | 0.628 | 13.01 | 4719 |
| NSRE | 0.417 | 0.341 | 0.493 | 10.80 | 4774 |
| SenRE | 0.416 | 0.338 | 0.494 | 10.47 | 4782 |

of moral duty to do so. Such effectiveness is also less vulnerable to future developments that replace ecosystem services or changes to people's perception of the importance of such services, since such changes do not affect nature's intrinsic values.

In this study, we chose to focus on the sense of moral duty towards the global natural environment, at a time scale of 50–100 years into the future, rather than the present, local environment. We believe that the scope chosen has high practical relevance. Today, neoliberal market favours the transfer of adverse environmental impacts from the developed to the developing region [40] through export, outsourcing and so on [41–43]. Climate change also imposes greater risks of temperature and rainfall extremes on the tropical region [44], while developed economies in Europe, North Asia and North America emit greenhouse gases in quantity disproportionate to their population. This means for many people with a high environmental footprint, their impacts are much more manifest outside their home region. Thus, a sense of duty towards *global* conservation is important so that those people may have a genuine concern for the damages happening far away from them. Second, psychological connectedness with a party may decrease with temporal distance [45], while the effects of environmental conservation usually have substantial time lags. For instance, even after greenhouse gas emission ceases, global temperature and sea level are set to rise for centuries if not millennium [46, 47]; a similar time lag in response to climate change is predicted for terrestrial ecosystems as well [48]. This effectively means a lot of conservation actions are more for the future Earth, and it is thus important for one to have a sense of moral duty directed at future entities despite the effect of temporal distance.

We acknowledge, however, that limitations come along with the above advantages for the scope chosen. One important trait of RV is that they may be rooted in people's cultural customs, daily practices and unique experiences of interacting with the local environment–in short, the situatedness of knowledge. The survey items and the moral judgment scenario are geared towards measuring global concern for the future, hence they may not measure well meanings attached to nature via unique experiences or concerns for the local environment. For instance, for some indigenous communities, the perception of nature is intertwined with cultural values and experiences linked to the local environment. Hence, they may not resonate with a general, abstract global conservation agenda not connected to their local knowledge and unique experiences. This does not make those people any less caring about the environment. The upshot of the discussion is that just like the sense of moral duty, the practical significance of recognising RV is also context-dependent. There may not be a one-size-fits-all type of moral values that explains the sense of moral duty towards nature conservation regardless of context.

A caveat of the above discussion is that the results should not be interpreted as a simple rejection of the RV concept. First, the failure to find support for H1 only suggests that the dependent relationship is not accounted for in people's moral judgment in this study. The notion that dependence is an essential element of RV is derived from our interpretation of the literature. The same applies to the survey items used in this study. We recognise that diverse interpretations of the RV concept may exist, and they may show different results when operationalised and measured. Second, as explained previously, this study adopted a weaker conception of RV. If one adopts the stronger conception, one may deny the possibility of the experimental condition (humans becoming independent from nature)–though this means one must commit to the bold empirical assumptions explained previously. Third, this study only concerns whether RV explain people's sense of duty for *future* nature, rather than the *present* nature. We thus do not deny the prospect that RV may explain well the sense of duty to prevent sudden, drastic changes to the present nature (especially locally) such as deforestation or destructive fishing practices, which destroy the ecosystem with which present humans have a morally significant relationship.

The assumptions made in "The Last Human" story may be extreme and sometimes unrealistic. Nevertheless, there are reasons to believe that this does not affect the reliability of the results. Firstly, the possible effects of people being unable to imagine humans becoming independent from nature has been controlled through the "unconscious realism" variable explained before. Secondly, empirical study found that folk intuition converges with philosophers' intuition on thought experiments, including ones that involve possibly hard-to-imagine scenarios (alternative universes that are causally deterministic/indeterministic) [49]. This implies that if philosophers do not find Routley's original last [hu]man argument hard to imagine, there is also no reason to question the respondents' ability to imagine and thus competently respond to "The Last Human" scenario.

The survey in this study was only done in Singapore. Although there is evidence that the patterns of people's ethical and metaethical intuitions are consistent when compared in different settings and that demographic traits have limited effects on the moral intuitions, we nevertheless acknowledge that ethical intuitions could vary for different communities and contexts, as would be expected from the diversity of human-nature relations. This potential diversity makes it essential that our work is replicated for different cultures, with especial importance that such replication occurs for the case of indigenous communities that are key stewards of remaining natural habitats.

One more point of note is that "dependence" in the survey of this study was not explicitly defined for conciseness. We recognise the possibility that respondents may have different interpretations of the term from each other. Future studies may provide detailed explanations and descriptions of the key concepts used in the survey.

## 7. Conclusion

We experimentally tested people's sense of moral duty towards global nature conservation for the future, as well as its correlation with people's recognition of the intrinsic/RV of various entities of nature. The results suggest how much people treasure the intrinsic values of sentient beings is the best predictor of such moral intuition. They do not suggest that Muracian RV explain such moral intuition. Our results thus suggest that intrinsic values of sentient beings may explain the sense of duty for global nature conservation for the future better than relational values do.

## Supporting information

**S1 Data.**
(XLSX)

## Acknowledgments

We would like to thank members of the NUS Bioecon Lab for providing ad-hoc advice on data analysis and logistical support to this study.

## Author Contributions

**Conceptualization:** Tianxiang Lan, Luis Roman Carrasco.

**Formal analysis:** Tianxiang Lan.

**Funding acquisition:** Luis Roman Carrasco.

**Methodology:** Tianxiang Lan, Luis Roman Carrasco.

**Supervision:** Neil Sinhababu, Luis Roman Carrasco.

**Writing – original draft:** Tianxiang Lan.

**Writing – review & editing:** Tianxiang Lan, Neil Sinhababu, Luis Roman Carrasco.

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
