## [Decision Letter · Decision Letter 0]

27 Jul 2022

PONE-D-22-16258For its own sake: intrinsic values explain the moral duty towards nature conservation better than relational valuesPLOS ONE

Dear Dr. Lan,

Thank you for submitting your manuscript to PLOS ONE. After careful consideration, we feel that it has merit but does not fully meet PLOS ONE’s publication criteria as it currently stands. Therefore, we invite you to submit a revised version of the manuscript that addresses the points raised during the review process.

We look forward to receiving your revised manuscript.

Kind regards,

Ingo Brigandt

Academic Editor

PLOS ONE

Journal Requirements:

2. Please provide additional details regarding participant consent. In the Methods section, please ensure that you have specified (1) whether consent was informed and (2) what type you obtained (for instance, written or verbal). If your study included minors, state whether you obtained consent from parents or guardians. If the need for consent was waived by the ethics committee, please include this information.

"YES - This research is supported by the National Research Foundation, Prime Minister’s Office, Singapore under its Campus for Research Excellence and Technological Enterprise (CREATE) Programme (NRF2016-ITC001-013)."

Additional Editor Comments:

I ask you to thoroughly consider the referees' concerns, especially the point by Referee 1 that the questions posed may not elicit the respective response. This may require doing new surveys or justifying the previously used questions while clarifying what issues the study does not speak to.

Reviewers' comments:

Reviewer's Responses to Questions

**Comments to the Author**

1. Is the manuscript technically sound, and do the data support the conclusions?

Reviewer #1: Partly

Reviewer #2: Yes

2. Has the statistical analysis been performed appropriately and rigorously? 

Reviewer #1: Yes

Reviewer #2: I Don't Know

3. Have the authors made all data underlying the findings in their manuscript fully available?

Reviewer #1: No

Reviewer #2: No

4. Is the manuscript presented in an intelligible fashion and written in standard English?

Reviewer #1: Yes

Reviewer #2: Yes

5. Review Comments to the Author

Reviewer #1: This article sets forth an empirical investigation into the theoretical ground for persons’ sense of duty toward the conversation of nature, and, related, identify the best theoretical predictors that persons hold a sense of duty to the conversation of nature.

As a general point, I’m pleased to see work in experimental philosophy targeting questions within environmental philosophy. Results from this type of work are worth generating and considering. Regrettably, though, I can’t recommend this paper for publication.

There are, as far as I can tell, no problems with the basic methodology and application of appropriate statistical tools. But the substance of the survey and the inferences drawn regarding the value of nature, I don’t think are successful.

The problem traces foundationally to the way in which the authors attempt to reveal respondents’ theoretic commitments to relational values. To be sure, the authors should be commended for detailing the Muracian framework, how they understand her accounting of relational values, as well as their acknowledging that questions of interpretation remain. Ultimately, though, I don’t think they’ve done enough in the set-up with their study to ensure that, IF respondents held something like a relational value view for conserving nature (or, perhaps, in keeping with the authors’ formulation, if relational values were psychological/emotional motivators for environmental protection), that their questions would demonstrate that.

Take first the authors’ version of the last person though experiment. As the original thought experiment intended, it’s perfectly reasonable to view those who judge it wrong for the last person to destroy the remaining living world to extend intrinsic value to that world. And we should expect to see consistency across this and the specific intrinsic value questions. But it’s not at all clear (in fact I’m skeptical) that either the last person thought experiment or the abstract questions regarding humans’ dependence on nature will prompt respondents to answer in a way reflective of their holding a relational value view of nature.

As I see it, the authors are asking too much of respondents by posing the questions as they do. For those persons who value in a way that is relational, the specific parts of nature and their unique connection to it is essential to making visible that valuing. A general question such as “I have a strong affinity towards nature and ecosystems which forms part of my sense of identity” strikes me as a question that many who may hold a relational value view of nature will not elicit a positive response. And the reason for this is that the respondent has to themselves recognize their own particular relational valuing from the abstract question.

Consider, for example, an avid and moderately enlightened duck hunter. This duck hunter has been hunting since childhood with friends and family. She has grown to rely on the duck for special occasion meals. She also has been a vocal proponent of wetlands protection to ensure the future of her being able to duck hunt. But when presented with the abstract question about having a strong affinity towards nature and ecosystems, the respondent has to work to make the connection between their relational valuing of wetlands and a positive response to that question. If, moreover, the duck hunter is contemptuous of most environmentalism (a very real possibility), their scoring of that question may be very low, despite them being a good example of a relational valuing of nature. The same, of course, would be true for the duck hunter’s treatment of the last person thought experiment. The implication, then, is that a failure to find respondents answering in ways consistent with a relational value framework may, plausibly, be a mirage; they do value nature on a relational basis, and thus seek nature conversation, but not GENERALLY as the question probes. They do it for their own particular parts of nature that they have a specific relationship with.

Notably, the authors gesture to concerns that relate to the above. First, they note that RV theorists highlight the context of valuing (p. 13). Second, they acknowledge that respondents may have difficulty understanding what relational values are (p. 15). For the first of these, the authors attempt controls for age and gender, where they find no statistical change. For the second, the authors suggest that probing for conservation along a temporal dimension would be useful. But neither of these points really address the essence of the problem outlined above.

In the controls for age and gender, it’s suggested that these would be proxies for the effects of context. But that doesn’t seem correct—see again the duck hunter case (and note that that is just a single illustration of the effect of one specific context). Regional and cultural factors are seemingly the more appropriate variables, but it’s important to keep in mind the issue centers on how to pose questions that optimize respondents’ ability to connect their particular human-natural world relationship with an inclination to protect the natural world (aside—as signaled above, I think there is also an interesting question that arises whether conservation of nature is derivatively local rather than global in a RV framework, again, countering the ability of the last person thought experiment to reveal respondents’ commitments to nature conservation within that framework). This basic point also plagues the temporal dimension idea for addressing respondents’ understanding of RV. The problem is not to ensure posing questions that attend to the temporal dimension of protecting nature; the worry is that the questions do not connect with the unique relationships respondents might have with parts of the natural world, showing them lacking relational values when in fact they have them and would seek (certain?) environmental protections were that connection made.

One small final point. The authors appear to use interchangeably the language of “explanatory” and “predictor.” Explanatory factors are (typically) temporally sensitive—the explanans come before the explanadum. For predictors, that need not be the case. In a circumstance like this, where certain statistical relationships are introduced to establish predictors, it’s probably not problematic to go further and assert they are explanatory. But that is an additional inference.

Reviewer #2: This paper engages with the question of whether relational values, that have in recent years come to be a point of discussion, in particular surrounding the work of IPBES that have adopted a framework based on this concept. The concept of relational values is supposed to to do something about various perceived weaknesses of conventional ways of thinking about the value of nature that rely on the distinction between intrinsic and instrumental values. What the authors of this paper strives to do is to test the extent to which relational values, as compared to intrinsic values, drive peoples’ reasons for thinking that (future) nature should be conserved. This is done experimentally by exposing a comparatively large group of people to two variations of the “last man” argument. This is designed so as to isolate the different kinds of values by subjecting subjects to different experimental conditions (in the aforementioned scenario). The authors conclude that it is intrinsic values rather than relational values that predict a sense of moral responsibility towards (future) nature.

The details of the statistical analysis I am not able to competently assess and will hence assume is sound.

My overall impression of this paper is that it is (as far as I can tell) a competently conducted study of the importance of relational values in predicting people’s moral attitudes towards nature and provides some evidence that appears to be in conflict with some earlier studies. It is mostly well written with a clear and easy to follow structure. A lot of the questions I found myself thinking about when reading this paper where subsequently brought up and dealt with in the discussion section. I think this paper is in a condition that is publishable as it is and the comments and remarks I have are minor.

Comments:

I think the main charge against this approach to relational values turns on how the concept is operationalised and tested at a very general level. The commitment to a weaker conception of relational values is convenient clearly and possibly is what allows for a study like this to be conducted but it seems to be based on assumptions that are antithetical to what many would believe to be significant about the turn to relational values. Especially the challenge to a sharp dichotomy between humans and nature is important as it would make a lot of the questions posed in this study appear somewhat meaningless. In the relational values framework, as I understand it, it just doesn’t make sense to talk of things like ‘dependence’ for example and the very idea of conservation becomes highly suspicious (and in practice often a vehicle for oppression). Now the authors flag an awareness of this but dismiss stronger varieties of the concept. Now, I am willing to accept the setup here. Testing stronger conceptions of relational values is perhaps more difficult to do rigorously.

The relational values are supposed to be removed from consideration in the experimental setup by presenting the subjects with a scenario where humans are not dependent on nature versus are dependent on nature and thus isolating the causal influence of relational values. I mostly wondering whether these prompts actually engage the right kind of thing in test. For example, how are respondents understanding 'dependence'? Is it possible to think that dependence is understood narrowly by respondents and that there might be other ‘non-dependence’ relationships that are the objects of the relational values and that the experimental setup is blind to these?

Regarding the concept of (personal) identity that is supposed to elicit from respondents relational values I am wondering how respondents understand these constructs. This is a way of operationalising the concept, of course, but it seems potentially narrow. I’m not entirely convinced from the discussion on relational values in the paper that this is an iron clad construction for getting at these values.

Sometimes the authors seem to conflate ‘conservation efforts’ with having certain moral attitudes towards nature. It is a minor thing, maybe, but I am not exactly sure these are the same necessarily. Conservation could perhaps be understood as something more narrow than simply caring about nature. I.e., one might think that conventional conservation does not adequately protect nature, or are even counter productive.

Page 3: Concerning the discussion on intrinsic versus instrumental values. I guess subjectivists about values will not think of intrinsic values as being ‘inherently possessed’ (maybe, I’m not sure what ‘inherently possessed’ is supposed to entail). For the subjectivist even intrinsic values typically requires a subject of some sort valuing the things (but doing so in a certain way).-

Page 4: The notion of ‘exploitative’ anthropocentrism is not clear to me. Why is it exploitative? In virtue of being anthropocentric? Or is it a kind of anthropocentrism that is exploitative? I suspect the former, but nonetheless. I’m not sure these qualifiers add to the clarity of the paper. Something similar could be said about ‘altruistic non-anthropocentrism.

Page 7: It seems to me that trying to investigate whether RV are “able” to drive a sense of moral duty towards nature is perhaps too strong. The framework might be the (normatively) right one, but still not be intuitive for many contemporaries. If they could be educated somehow this might change. Minimally it seems one could approach the relational values framework as a framework that has normative force (although not necessarily being the intuitive one for people in general) or as (also) having some descriptive content. I.e., it better reflects how people think.

P16 (line 319 and onwards). I’m not sure what the authors are trying to say here. They at least come very close to transgressing the line between the normative and descriptive. The results show that intrinsic values are enough to provide people (in the survey) with reasons for protecting nature. Or at least a sense of moral responsibility towards nature. It does not seem to adjudicate between the philosophical positions as philosophical positions. Although it might, as the authors seem to say, provide some idea of what kind of arguments might be persuasive vis a vis the general public.

To conclude, I really liked this papper and I think it is a nice and well-written contribution to the discussion on relational values and well deserves to be published. With some minor adjustments and clarifications the paper could be slightly improved.

6. PLOS authors have the option to publish the peer review history of their article (what does this mean?). If published, this will include your full peer review and any attached files.

Reviewer #1: No

Reviewer #2: No

---

## [Author Response · Author response to Decision Letter 0]

6 Sep 2022

Please see attached document "Response to reviewers" where we provided detailed replies to all the comments, thank you.

---

## [Decision Letter · Decision Letter 1]

11 Oct 2022

Recognition of intrinsic values of sentient beings explains the sense of moral duty towards global nature conservation

PONE-D-22-16258R1

Dear Dr. Lan,

We’re pleased to inform you that your manuscript has been judged scientifically suitable for publication and will be formally accepted for publication once it meets all outstanding technical requirements. But see the below PLOS Data policy.

Kind regards,

Ingo Brigandt

Academic Editor

PLOS ONE

Additional Editor Comments (optional):

Reiterating a point made by Referee 2, I draw you attention to the journal's data sharing policy, which calls for making accessible more of the data than summary statistics.

Reviewers' comments:

Reviewer's Responses to Questions

**Comments to the Author**

1. If the authors have adequately addressed your comments raised in a previous round of review and you feel that this manuscript is now acceptable for publication, you may indicate that here to bypass the “Comments to the Author” section, enter your conflict of interest statement in the “Confidential to Editor” section, and submit your "Accept" recommendation.

Reviewer #1: All comments have been addressed

Reviewer #2: All comments have been addressed

2. Is the manuscript technically sound, and do the data support the conclusions?

Reviewer #1: Yes

Reviewer #2: Yes

3. Has the statistical analysis been performed appropriately and rigorously? 

Reviewer #1: I Don't Know

Reviewer #2: Yes

4. Have the authors made all data underlying the findings in their manuscript fully available?

Reviewer #1: Yes

Reviewer #2: No

5. Is the manuscript presented in an intelligible fashion and written in standard English?

Reviewer #1: Yes

Reviewer #2: Yes

6. Review Comments to the Author

Reviewer #1: The authors do a reasonably good job addressing the comments made in my previous review. My sense is there remains a good bit to critically analyze regarding both the philosophical lessons, and, more importantly perhaps, the relevance for conservation efforts. But the survey results and interpretation of them provide some opportunity to do that analysis.

Reviewer #2: I think the authors have done a good job of addressing the comments and I'm happy to recommend this article for publication.

7. PLOS authors have the option to publish the peer review history of their article (what does this mean?). If published, this will include your full peer review and any attached files.

Reviewer #1: No

Reviewer #2: No

---

## [Editor Report · Acceptance letter]

18 Oct 2022

PONE-D-22-16258R1 

Recognition of intrinsic values of sentient beings explains the sense of moral duty towards global nature conservation 

Dear Dr. Lan:

I'm pleased to inform you that your manuscript has been deemed suitable for publication in PLOS ONE. Congratulations! Your manuscript is now with our production department. 

Kind regards, 

on behalf of

Dr. Ingo Brigandt 

Academic Editor

PLOS ONE